biochemistry

spin labelling, amyloid, protein aggregation, yeast prion, electron paramagnetic resonance

**Author for correspondence:**
Zhefeng Guo
e-mail: zhefeng@ucla.edu

# Effect of spin labelling on the aggregation kinetics of yeast prion protein Ure2

Emilie N. Liu, Giovanna Park, Junsuke Nohara and Zhefeng Guo

Department of Neurology, Brain Research Institute, Molecular Biology Institute, University of California, Los Angeles, CA 90095, USA

ZG, 0000-0003-1992-7255

Amyloid formation is involved in a wide range of neurodegenerative diseases including Alzheimer's and prion diseases. Structural understanding of the amyloid is critical to delineate the mechanism of aggregation and its pathological spreading. Site-directed spin labelling has emerged as a powerful structural tool in the studies of amyloid structures and provided structural evidence for the parallel in-register β-sheet structure for a wide range of amyloid proteins. It is generally accepted that spin labelling does not disrupt the structure of the amyloid fibrils, the end product of protein aggregation. The effect on the rate of protein aggregation, however, has not been well characterized. Here, we employed a scanning mutagenesis approach to study the effect of spin labelling on the aggregation rate of 79 spin-labelled variants of the Ure2 prion domain. The aggregation of Ure2 protein is the basis of yeast prion [URE3]. We found that all spin-labelled Ure2 mutants aggregated within the experimental timeframe of 15 to 40 h. Among the 79 spin-labelled positions, only five residue sites (N23, N27, S33, I35 and G42) showed a dramatic delay in the aggregation rate as a result of spin labelling. These positions may be important for fibril nucleation, a rate-limiting step in aggregation. Importantly, spin labelling at most of the sites had a muted effect on Ure2 aggregation kinetics, showing a general tolerance of spin labelling in protein aggregation studies.

## 1. Introduction

Protein aggregation, a supersaturation-driven process [1,2], and formation of amyloid fibrils are the basis of a wide range of human disorders such as Alzheimer's, Parkinson's and prion diseases [3–5]. Structural studies of these protein aggregates are important for delineating the mechanism of protein aggregation underlying the pathogenesis of amyloid-related disorders.

Site-directed spin labelling in combination with electron paramagnetic resonance (EPR) spectroscopy [6,7] has proven to be a powerful tool in the structural studies of amyloid fibrils and soluble oligomers. In the early years of amyloid structural studies, spin labelling EPR [8], along with solid-state NMR [9], provided crucial evidence for the parallel in-register β-sheet structure that is characteristic of most amyloid fibrils. Proteins whose aggregation and amyloid structures have been studied with EPR include Aβ [10–15], tau [16–18], α-synuclein [19,20], human prion [21], islet amyloid polypeptide [22,23], β2-microglobulin [24], huntingtin [25,26], transthyretin [27], Orb2 [28], HET-s [29], Sup35 [30] and Ure2 [31–34]. Structures of Aβ oligomers have also been studied with EPR [35–39]. Studies from hundreds of spin-labelled amyloid protein variants showed that spin labelling in general does not disrupt the structures of these amyloid aggregates. One area that is still not well characterized, however, is how spin labelling affects the rate of protein aggregation. Here, we address this question by studying the aggregation rate of 79 spin-labelled variants of yeast prion protein Ure2.

Ure2 protein aggregation is required for the formation of the [URE3] prion in *Saccharomyces cerevisiae* [40]. The full-length Ure2 protein is 354 residues long and consists of two domains. The N-terminal prion domain (approx. 90 residues) is required for the [URE3] prion phenotype and the C-terminal functional domain is necessary and sufficient for the cellular function of Ure2 [41,42]. In its normal role, Ure2 protein is responsible for the suppression of protein expression involved in the uptake of poor nitrogen sources when a good nitrogen source is present. However, in the prion state of [URE3], the aggregation of the Ure2 prion domain sequesters the C-terminal functional domain, resulting in the expression of genes responsible for the catabolism of poor nitrogen sources even in the presence of a good nitrogen source. Yeast prions share many properties, such as infectivity and the prion strain phenomenon, with human prions, making yeast prions a good model system for the studies of human prion and amyloid-related diseases. Ure2 prion domain is necessary for the prion phenomenon *in vivo* and sufficient for the formation of amyloid fibrils *in vitro* [43,44]. Tycko and co-workers [45] used solid-state NMR to show that the Ure2 prion domain adopts a parallel in-register β-sheet structure in fibrils. They further showed that the Ure2 prion domain forms the amyloid core of the full-length Ure2 fibrils [46]. Cysteine scanning mutagenesis has been used to identify key residues for the fibrillization process of Ure2 [47], and the mutation R17C was identified to greatly accelerate the aggregation rate of Ure2 under oxidizing conditions.

Using site-directed spin labelling and EPR, we have previously shown that Ure2 adopts a parallel in-register β-sheet structure in the fibrils formed by the prion domain [31]. By immobilizing Ure2 protein on a solid-support, we demonstrated that monomeric Ure2 prion domain is completely disordered [32]. This is consistent with previous studies that show the disordered nature of Ure2 prion domain [48], suggesting that spin labelling does not affect the conformation of soluble Ure2. Quantitative analysis allowed us to identify likely β-strand regions in the Ure2 fibrils [33,34]. Through these studies, we have obtained a complete library of spin-labelled variants spanning residues 2–80 of the Ure2 prion domain. This allowed us to perform a comprehensive study of the effect of spin labelling on the aggregation kinetics of the Ure2 prion domain. Our studies show that spin labelling does not dramatically affect the aggregation rate at all but five residue positions, which may represent some key residues in the fibril nucleation, a rate-limiting step of protein aggregation.

# 2. Results and discussion

We prepared 79 spin-labelled mutants with spin labels introduced, one at a time, at residues 2–80 of the Ure2 prion domain using site-directed spin labelling, which includes site-directed mutagenesis of a residue to cysteine and subsequent covalent labelling to attach the spin label side chain named R1 (figure 1). Previously, we have prepared amyloid fibrils using all these spin-labelled mutants and performed EPR studies [31,33,34]. For fibrillization kinetics experiments, the lyophilized Ure2 protein was first dissolved in a denaturing buffer containing 7 M guanidine hydrochloride, and then diluted 20-fold to PBS to initiate aggregation. All aggregation experiments were performed at 37°C without agitation, lasting approximately 15 to 40 h depending on different samples. The aggregation process was monitored using thioflavin T fluorescence, which is a quantitative measurement of amyloid formation for fibrils of the same structure [49]. The aggregation kinetics of wild-type Ure2 prion domain is shown in figure 2*a*. Fibril morphology was observed at the end of the kinetics experiment, as shown in figure 2*b*. The aggregation data of all the spin-labelled Ure2 mutants are shown in figures 3 and 4. Within the experimental timeframe in this work, all of the spin-labelled mutants had the ability to aggregate and formed amyloid fibrils, demonstrating that the spin label did not prevent

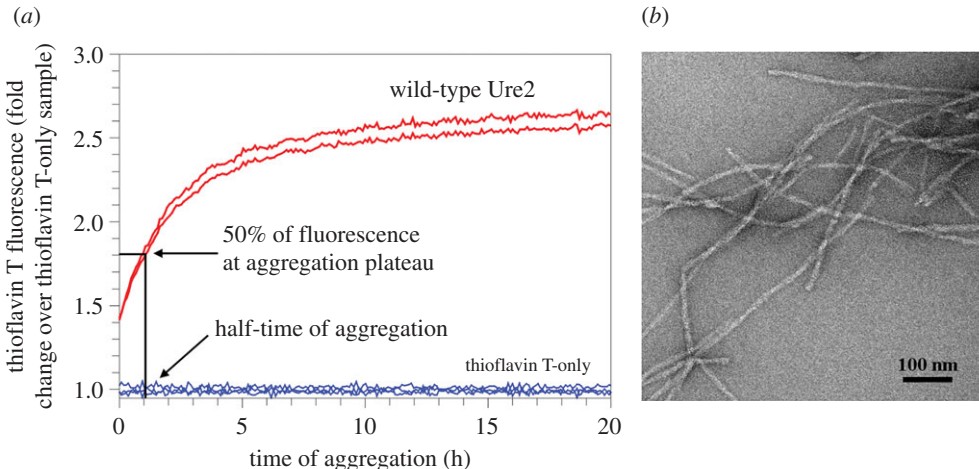

(a)

protein-SH +

spin label MTSSL

(b)

protein

spin label side chain R1

**Figure 1.** Spin labelling reaction to attach the commonly used spin label side chain R1.

(a)

wild-type Ure2

50% of fluorescence
at aggregation plateau

half-time of aggregation

thioflavin T-only

(b)

100 nm

**Figure 2.** Aggregation of wild-type Ure2 prion domain. (a) Aggregation kinetics of Ure2 prion domain monitored with thioflavin T fluorescence. The aggregation was performed using 10 µM Ure2 in PBS buffer (pH 7.4, containing 0.35 M guanidine hydrochloride) at 37°C without agitation. The half-time of aggregation, the time to reach 50% of the fluorescence intensity at aggregation plateau, is determined directly from the kinetics data. (b) Transmission electron micrograph of Ure2 aggregates at the end of kinetics experiment.

aggregation when introduced to any of the residue positions. The vast majority of mutants aggregated quickly without a detectable lag phase. To quantitatively evaluate the aggregate rate, we determined the half-time of aggregation, which is the time to reach 50% of the fluorescence at the aggregation plateau (figure 2a). For most mutants, we used the background fluorescence of thioflavin T sample as the starting fluorescence due to the lack of a lag phase, in which fibril nuclei are formed but thioflavin T fluorescence remains near background level. If the fluorescence value at the beginning of the aggregation was already more than 50% of the value at aggregation plateau, then we arbitrarily set the half-time at zero. The half-time values as a function of residue positions of Ure2 prion domain are plotted in figure 5.

As shown in figure 5, the half-time values for most mutants are clustered in a range of 0–5 h. Approximately half of the mutants (43 out of 79) aggregated faster than wild-type Ure2, which has a half-time of 1.0 h, while the other half of the mutants aggregated slower. The aggregation data collectively suggest that spin labelling does not dramatically affect the aggregation rate at most of the residue positions. One notable trend is that spin labelling at the N-terminal half of the prion domain mostly delayed aggregation. For residues 2–40, spin labelling at 67% of the residues positions showed half-time longer than wild-type. Spin labelling at charged residues overall has inhibitory effect on aggregation, with six out of eight charged residue positions showing larger half-time values than wild-type.

Spin labelling at five residue positions, N23, N27, S33, I35 and G42, showed a dramatic decrease in aggregation rate. With wild-type and most other mutants showing half-time between 0 and 5 h, these five Ure2 mutants gave much larger half-time values at 26.2 h (N23R1), 27.3 h (N27R1), 13.1 h (S33R1), 23.5 h (I35R1) and 14.0 h (G42R1). The half-time numbers for these five mutants are more than two standard deviations larger than the average of all the mutants analysed. These five mutants are also the only mutants in this study that show prolonged lag phases. Our previous EPR studies on the spin-labelled Ure2 fibrils show that spin labelling did not disrupt the parallel in-register β-sheet structure [31,33,34]. Among these five residues, only G42R1 shows a lack of β-sheet structure in fibrils (figure 6).

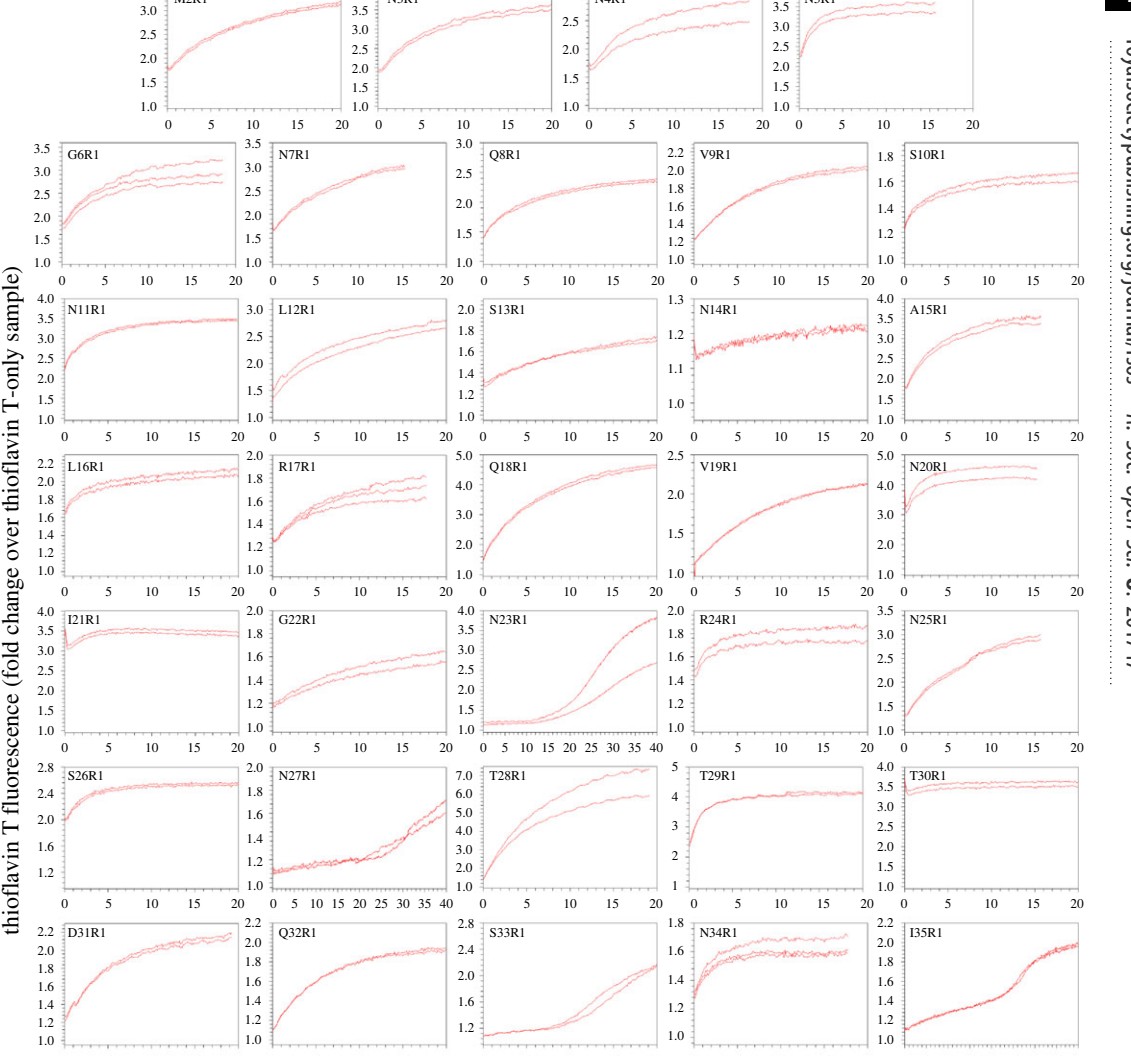

**Figure 3.** Aggregation kinetics of spin-labelled Ure2 mutants at positions 2–40. R1 represents the spin label. All aggregations were performed using 10 μM Ure2 in PBS buffer (pH 7.4, containing 0.35 M guanidine hydrochloride) at 37°C without agitation.

Therefore, the inhibition of aggregation rate at these residue positions does not appear to cause disruption of β-sheet structure overall.

Further analysis of the aggregation data of 79 Ure2 mutants suggests that site-specific effects on the rate of aggregation do not correlate with changes in thioflavin T fluorescence at aggregation plateau. Figure 7 shows a plot of thioflavin T fluorescence at aggregation plateau versus the half-time of aggregation for all the Ure2 mutants. It has been shown that thioflavin T fluorescence is a quantitative measure of fibril amount for the same fibril structure [49]. In mutagenesis studies of amyloid proteins, reduction in thioflavin T fluorescence has often been interpreted as inhibition of aggregation. A mutation can reduce the aggregation yield, but can also change the structure of the aggregate, which may bind thioflavin T with different affinity or give different fluorescence quantum yield even with the same binding affinity. Therefore, when comparing mutants and wild-type aggregations, thioflavin T fluorescence intensity by itself cannot be used to quantify the effect of mutations on aggregation. We reached the same conclusion in a previous study of Aβ42, in which we performed kinetics studies

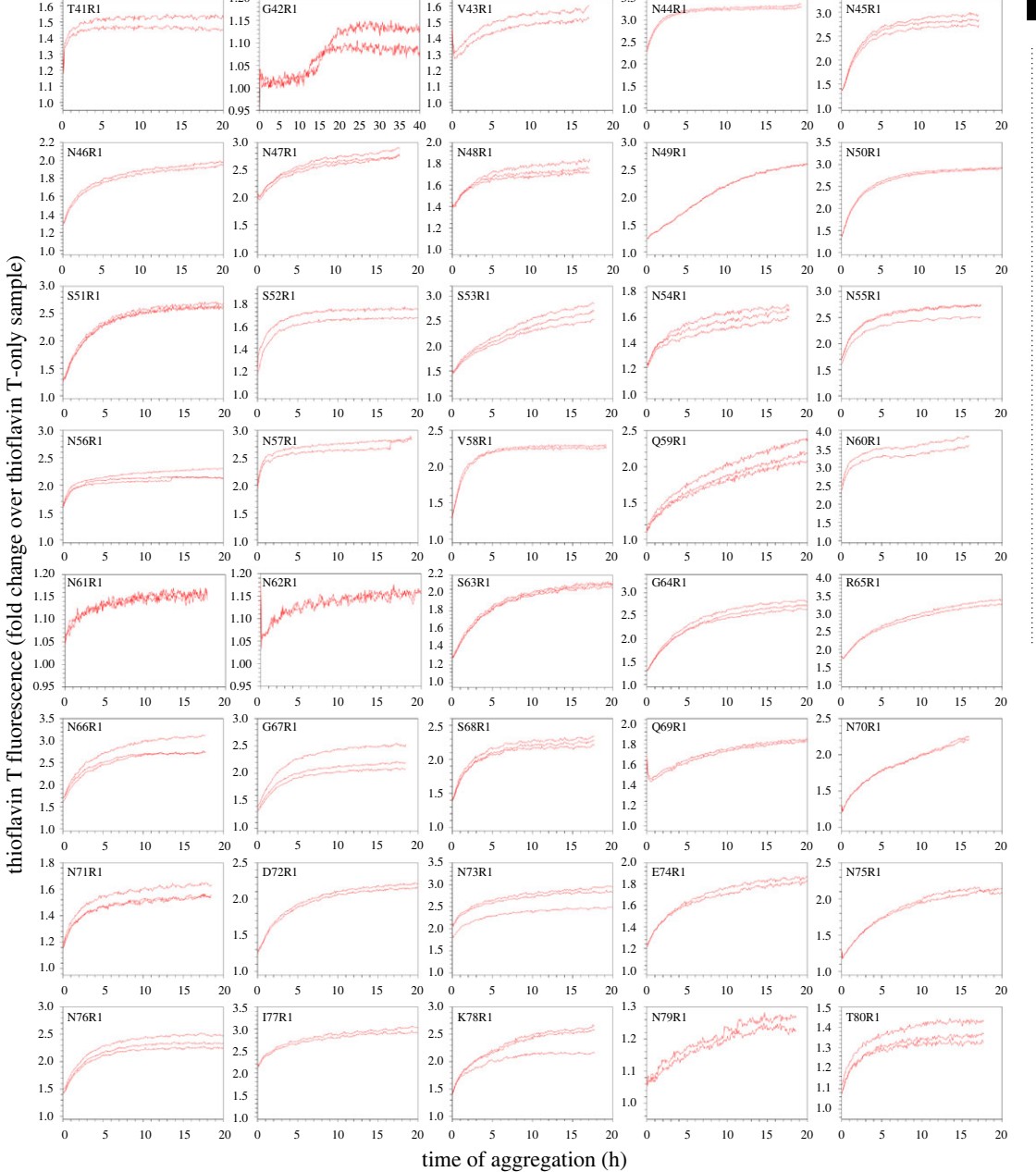

**Figure 4.** Aggregation kinetics of spin-labelled Ure2 mutants at positions 41–80. R1 represents the spin label. All aggregations were performed using 10 μM Ure2 in PBS buffer (pH 7.4, containing 0.35 M guanidine hydrochloride) at 37°C without agitation.

of 42 spin-labelled Aβ42 mutants and found that the half-time of aggregation did not correlate with thioflavin T fluorescence at aggregation plateau [50].

To help understand the aggregation data, we colour-coded the residues in a schematic of Ure2 sequence that includes secondary structure information from our previous EPR studies (figure 8). Out of the five most inhibiting residues, N23 and G42 are located at turn regions. S33 is located on a β-strand, but right adjacent to a turn. N27 and I35 are located in the middle of β-strands. The analysis suggests that the formation of turns is as important as β-strands in the nucleation of fibrils, a key rate-limiting step in protein aggregation. Similarly, our previous kinetics studies of 42 spin-labelled Aβ42 mutants showed that mutations at a turn or loop sites had the most dramatic effect on Aβ42 aggregation [50]. However, not all mutations at turns had the same effect, suggesting that there are site-specific interactions involved in fibril nucleation. Residue positions with the most inhibiting effect do not appear to have a preference on residue type. These five residue positions include two glutamines, one isoleucine, one serine and one glycine. Notably, the five residues do not have charged amino acids, even though there are eight charged residues in the region of 2–80.

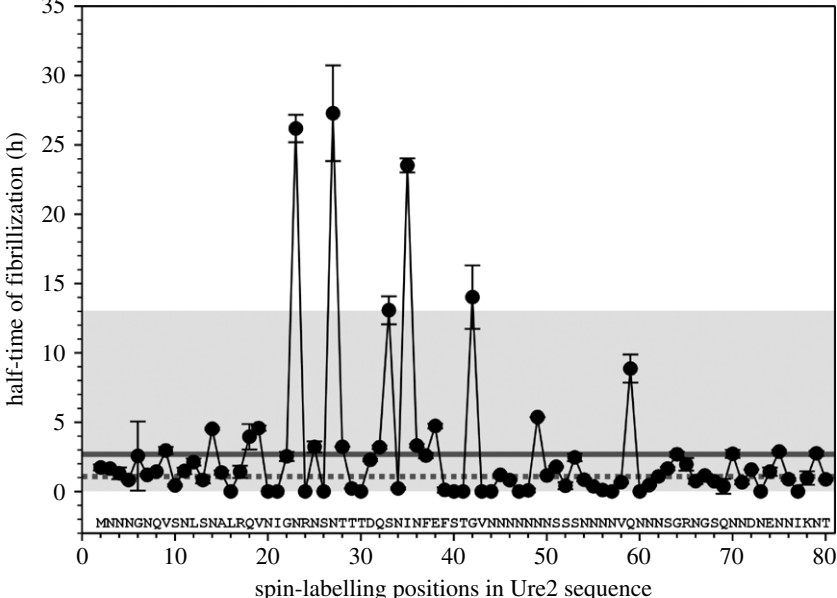

**Figure 5.** Plot of half-time of aggregation versus residue positions in Ure2 sequence. Half-time is defined as the time taken to reach 50% of the thioflavin T fluorescence at aggregation plateau. Each data point is the average of either duplicates or triplicates for each Ure2 mutant, as shown in figures 3 and 4, and error bars are the standard deviation. The dotted and straight lines represent the half-time values for wild-type Ure2 and the average of all mutants, respectively. The shaded box shows the range within two standard deviations from the average of all data points. Note that only five mutants show half-time values that are more than two standard deviations from the average.

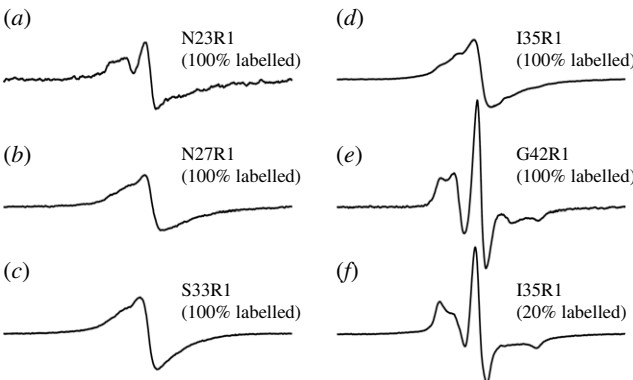

**Figure 6.** EPR spectra of the Ure2 fibrils spin-labelled at positions 23, 27, 33, 35 and 42. (a–d) Fibrils prepared with only spin-labelled Ure2 mutants, N23R1 (a), N27R1 (b), S33R1 (c) and I35R1 (d), show characteristic single-line EPR feature, as a result of molecular stacking of spin labels in a parallel in-register β-sheet structure. (e) The EPR spectrum of Ure2 G42R1 variant shows a typical three-line feature, suggesting an absence of parallel β-sheet packing at position 42. (f) The EPR spectrum of the mixture of I35R1 and wild-type Ure2 at 1 : 4 molar ratio (i.e. 20% labelled) shows a typical three-line feature due to diminished spin label stacking. The EPR spectra of 100% labelled N23R1, N27R1, S33R1, I35R1 and G42R1 are reproduced using data in Wang et al. [34]. The EPR spectrum of 20% labelled I35R1 is reproduced using data in Ngo et al. [31].

Previously, Fei & Perrett [47] identified that the R17C mutation accelerated Ure2 aggregation under oxidizing conditions and the peptide corresponding to the sequence of residues 17–24 can form amyloid fibrils *in vitro*. Deletion of residues 15–41 dramatically reduced the rate of Ure2 aggregation *in vitro* [51]. Ventura and co-workers [52] used the pWALTZ algorithm to predict and then experimentally validated that the 21-residue stretch spanning residues 20 to 40 of Ure2 formed an amyloid core. Hydrogen exchange studies by Melki and co-workers [53] show that fibrillization gave the highest level of protection to the amide protons of residues 13–37. Solid-state NMR studies by Tycko and co-workers [54] show that the peptide corresponding to residues 10–39 of the Ure2 prion domain formed fibrils with parallel in-register structure. Edskes & Wickner [55] show that residues 2–44 are important for prion induction and curing *in vivo*. These studies support the key structural role of the N-terminal

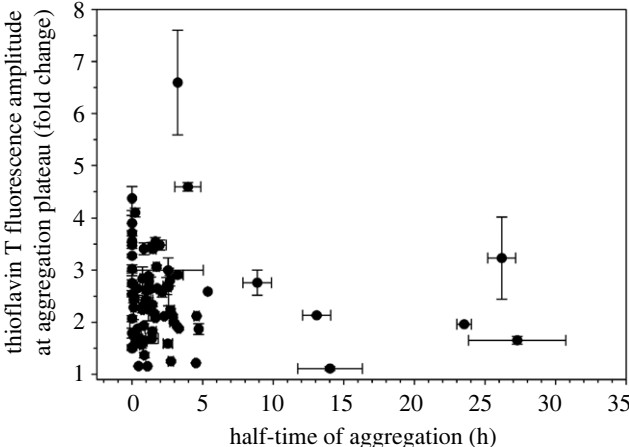

**Figure 7.** Plot of thioflavin T fluorescence at aggregation plateau versus half-time values for all spin-labelled Ure2 mutants. Note that there is no correlation between half-time of aggregation and the thioflavin T fluorescence at aggregation plateau.

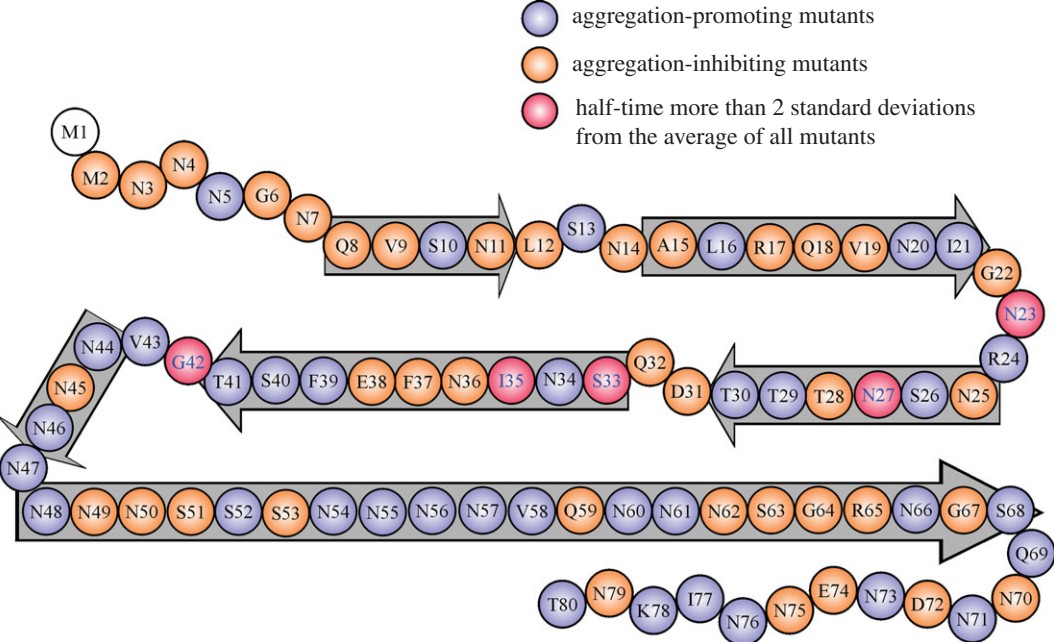

**Figure 8.** Ure2 prion domain sequence with colour coding according to the inhibiting or promoting effect of spin labelling. Aggregation-promoting mutants with shorter half-time than wild-type are shown in blue. Aggregation-inhibiting mutants with longer half-time than wild-type are shown in orange. The five mutants with half-time more than two standard deviations from the average of all mutants are shown in magenta. Block arrows represent likely β-strands determined from EPR data in Wang *et al.* [34]. Note that residues 48–68 are shown as a continuous β-strand based solely on EPR data and probably consist of at least one turn region in the middle.

half of the Ure2 prion domain in the formation of Ure2 fibrils. Our findings that the five residues with the most dramatic effect on Ure2 aggregation rate are clustered between residues 23 and 42 suggest that this region is also kinetically important for the aggregation of Ure2 prion domain.

## 3. Conclusion

Our results suggest that spin labelling in general has a muted effect on the aggregation of Ure2. Only 5 out of 79 spin-labelled Ure2 variants showed markedly reduced aggregate rate. The clustering of these five residues in the region of 23–42 is consistent with previous findings that support the key structural role of the N-terminal half of the Ure2 prion domain. Interestingly, spin labelling at six out of eight

charged residue positions inhibited the rate of Ure2 aggregation, raising the prospect that the charged residues may serve as a molecular lubricant to provide necessary structural flexibility for fibril nucleation.

# 4. Material and methods

## 4.1. Ure2 protein preparation and spin labelling

The construct of Ure2 protein in this work contains the Ure2p prion domain (residues 1–89) and the M domain (residues 125–253) of yeast prion protein Sup35p fused at the C-terminus. The full sequence of this construct was described in Alberti *et al.* [56]. Cysteine mutants were introduced individually at residue positions 2–80 of the Ure2p prion domain using the QuikChange site-directed mutagenesis kit (Agilent) [31,34]. Protein expression, purification and spin labelling were performed as previously described [31,34]. Finally, spin-labelled Ure2 proteins were buffer-exchanged to 30 mM $NH_4$ acetate (pH 10) to remove the excess free spin label, lyophilized and stored at $-80°C$.

## 4.2. Aggregation kinetics

The lyophilized Ure2 samples (wild-type and 79 spin-labelled mutants) were dissolved in PG buffer (15 mM sodium phosphate, 7 M guanidine hydrochloride, pH 6.8) to a concentration of 200 µM. Then, 2.5 µl of each sample was mixed with 42.5 µl of PBS (50 mM phosphate, 140 mM NaCl, pH 7.4) and 5 µl of thioflavin T stock solution (200 µM in PBS buffer). In the final aggregation mixture, the Ure2p concentration is 10 µM and thioflavin T concentration is 20 µM. For each mutant, either duplicates or triplicates were run depending on sample availability. The sample of 50 µl in total volume was then loaded into a black 384-well non-binding surface microplate with a clear bottom (Corning product 3655) and sealed with a polyester-based sealing film (Corning product PCR-SP). Fluorescence was measured from the bottom of the plate using an excitation filter of 450 nm and an emission filter of 490 nm in a Victor 3 V plate reader (Perkin Elmer). All aggregation experiments were performed at 37°C without agitation. The data are presented in fold change of thioflavin T fluorescence calculated by dividing the average fluorescence intensity of the thioflavin T only sample at each time point.

## 4.3. Transmission electron microscopy

Aggregated wild-type Ure2 prion domain sample (5 µl) at the end of the kinetics experiment was placed on glow-discharged copper grids covered with 400-mesh Formvar/carbon film (Ted Pella). Then, the samples were negatively stained using 2% uranyl acetate and examined under a FEI T12 electron microscope with an accelerating voltage of 120 kV.

## 4.4. Data analysis

The rate of fibrillization was quantified using the parameter half-time, which is defined as the time at which the thioflavin T fluorescence reaches 50% of the fluorescence at the aggregation plateau. For mutants that did not appear to reach a plateau during the time measured, the maximum amplitude at the end of aggregation was used instead. The measurement is determined directly from the kinetics curves without fitting. For mutants that started aggregation with thioflavin T fluorescence already more than 50% of the plateau values, the half-time is arbitrarily set at zero.

Data accessibility. Data available in the electronic supplementary material.
Authors' contributions. E.N.L., G.P. and J.N. designed and carried out experiments, analysed data and drafted the manuscript. Z.G. conceived and supervised the study, designed the experiments and drafted the manuscript.
Competing interests. The authors declare no competing interests.
Funding. Funding was provided by the National Institutes of Health (grant no. R01GM110448).
Acknowledgements. We thank the members of the Guo laboratory for technical assistance and discussions.

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
