## [Peer Review File · Royal Society Open Science]

Review History

RSOS-201747.R0 (Original submission)

Review form: Reviewer 1

Is the manuscript scientifically sound in its present form?

Yes

Are the interpretations and conclusions justified by the results?

Yes

Is the language acceptable?

Yes

Do you have any ethical concerns with this paper?

No

Have you any concerns about statistical analyses in this paper?

Yes

Recommendation?

Accept with minor revision (please list in comments)

Comments to the Author(s)

See attached file for comments (Appendix A).

Review form: Reviewer 2**Is the manuscript scientifically sound in its present form?**

Yes

Are the interpretations and conclusions justified by the results?

Yes

Is the language acceptable?

Yes

Do you have any ethical concerns with this paper?

No

Have you any concerns about statistical analyses in this paper?

No

Recommendation?

Accept with minor revision (please list in comments)

Comments to the Author(s)

In this manuscript, Liu and colleagues investigated how spine labelling on different regions of Ure2 affects its aggregation kinetics. The authors did a huge amount of work in preparing 79 spin-labelled Ure2 variants, respectively, which enables systematic comparison of all the variants of their aggregation kinetics. Overall, the work is technically sound, and is interesting and of importance to understanding the distinctive role of each individual residue on Ure2 in its fibrillation. I only have a few comments the authors might need to address in order to strengthen the manuscript.

1. The author might need to provide an EM image of at least the wild-type Ure2 sample in the end of the ThT kinetics assay. I am wondering whether Ure2 forms typical fibrils after incubation and how it looks like.
2. Does the spin-labelled influence the conformation of soluble Ure2?
3. The author might need to discuss further on Figure 6 in combined with previous biochemical and cellular results. Whether the finding in this study can explain the previous observation for Ure2 aggregation and function in yeast.

Decision letter (RSOS-201747.R0)

Dear Dr Guo

On behalf of the Editors, we are pleased to inform you that your Manuscript RSOS-201747 "Effect of spin labeling on the aggregation kinetics of yeast prion protein Ure2" has been accepted for publication in Royal Society Open Science subject to minor revision in accordance with the referees' reports. Please find the referees' comments along with any feedback from the Editors below my signature.

Please submit your revised manuscript and required files (see below) no later than 7 days from today's (ie 03-Feb-2021) date. Note: the ScholarOne system will 'lock' if submission of the revision is attempted 7 or more days after the deadline. If you do not think you will be able to meet this deadline please contact the editorial office immediately.

on behalf of Professor Luning Liu (Associate Editor) and Malcolm White (Subject Editor)
openscience@royalsociety.org

Reviewer comments to Author:
Reviewer: 1

Comments to the Author(s)

The authors present a simple but comprehensive study comparing the effect of mutant proteins previously used to determine structural information by EPR on aggregation using ThT fluorescence. While it is a simple study the results are important for understanding of how spin labels may affect aggregation kinetics of Ure2 which could be important for future studies as suggested. I have some specific comments I feel should be addressed prior to publication of this manuscript to improve the relation of current findings to already known literature:

The article needs a conclusion section

The authors state in the introduction that R17C accelerates Ure2 under oxidising conditions implying this is a critical residue but then do not mention in the discussion how or if this relates to their current data.

The buffer in the ThT reaction still contains a proportion of GdHCl (350mM) which should be stated and included in the figure legends.

Figure 4 would benefit from the mean being shown.

The authors discuss a dramatic difference, normally differences are discussed with relation to significance. I feel a specialist statistical reviewer should confirm if the statistical approach of taking average and 2SD (using n=2 or 3 per condition) is appropriate for this data rather than a more standard approach of comparison to WT protein values.

In the first paragraph of the results the authors refer to ThT as a quantitative measurement of amyloid fibrils but then later contradict this saying that ThT cannot be used as a quantitative measurement, this disparity should be removed.

More details and discussion regarding the relevance of G42R1 being the only mutant to show lack of beta-sheet in previous EPR should be included and a discussion as to why the others may not cause a structural alteration with relation to currently presented findings.

The authors comment on mutant positions that are near a turn (23, 42 and 33) however on their schematic position 33 appears to be within a beta-sheet region. This section should be re-written to fit with the schematic and further discussion of the implications of turn versus sheet positions.

Reviewer: 2

Comments to the Author(s)

In this manuscript, Liu and colleagues investigated how spine labelling on different regions of Ure2 effects its aggregation kinetics. The authors did a huge amount of work in preparing 79 spin-labelled Ure2 variants, respectively, which enables systematically comparison of all the variants of their aggregation kinetics. Overall, the work is technically sound, and is interesting and of importance to understanding the distinctive role of each individual residues on Ure2 in its fibrillation. I only have a few comments the authors might need to address in order to strengthen the manuscript.

1. The authors might need to provide EM images of at least the wild-type Ure2 sample in the end of the ThT kinetics assay. I am wondering whether Ure2 forms typical fibrils after incubation and what it looks like.
2. Does the spin-label influence the conformation of soluble Ure2?
3. The author might wish to discuss further whether the findings in this study can explain the previous observations for Ure2 aggregation and function in yeast?

===PREPARING YOUR MANUSCRIPT===

===PREPARING YOUR REVISION IN SCHOLARONE===

<https://royalsociety.org/journals/authors/author-guidelines/#supplementary-material> to include a suitable title and informative caption. An example of appropriate titling and captioning may be found at https://figshare.com/articles/Table_S2_from_Is_there_a_trade-off_between_peak_performance_and_performance_breadth_across_temperatures_for_aerobic_sc_ope_in_teleost_fishes_/3843624.

Author's Response to Decision Letter for (RSOS-201747.R0)

See Appendix B.

Decision letter (RSOS-201747.R1)

Dear Dr Guo,

It is a pleasure to accept your manuscript entitled "Effect of spin labeling on the aggregation kinetics of yeast prion protein Ure2" in its current form for publication in Royal Society Open Science.

Due to rapid publication and an extremely tight schedule, if comments are not received, your paper may experience a delay in publication. Royal Society Open Science operates under a continuous publication model. Your article will be published straight into the next open issue and this will be the final version of the paper. As such, it can be cited immediately by other

researchers. As the issue version of your paper will be the only version to be published I would advise you to check your proofs thoroughly as changes cannot be made once the paper is published.

on behalf of Professor Luning Liu (Associate Editor) and Malcolm White (Subject Editor)
openscience@royalsociety.org

Appendix A

The authors present a simple but comprehensive study comparing the effect of mutant proteins previously used to determine structural information by EPR on aggregation using ThT fluorescence. While it is a simple study the results are important for understanding of how spin labels may affect aggregation kinetics of Ure2 which could be important for future studies as suggested. I have some specific comments I feel should be addressed prior to publication of this manuscript to improve the relation of current findings to already known literature:

1. The article needs a conclusion section
2. The authors state in the introduction that R17C accelerates Ure2 under oxidising conditions implying this is a critical residue but then do not mention in the discussion how or if this relates to their current data.
3. The buffer in the ThT reaction still contains a proportion of GdHCl (350mM) which should be stated and included in the figure legends.
4. Figure 4 would benefit from the mean being shown.
5. The authors discuss a dramatic difference, normally differences are discussed with relation to significance. I feel a specialist statistical reviewer should confirm if the statistical approach of taking average and 2SD (using n=2 or 3 per condition) is appropriate for this data rather than a more standard approach of comparison to WT protein values.
6. In the first paragraph of the results the authors refer to ThT as a quantitative measurement of amyloid fibrils but then later contradict this saying that ThT cannot be used as a quantitative measurement, this disparity should be removed.
7. More details and discussion regarding the relevance of G42R1 being the only mutant to show lack of beta-sheet in previous EPR should be included and a discussion as to why the others may not cause a structural alteration with relation to currently presented findings.
8. The authors comment on mutant positions that are near a turn (23, 42 and 33) however on their schematic position 33 appears to be within a beta-sheet region. This section should be re-written to fit with the schematic and further discussion of the implications of turn versus sheet positions.

Appendix B

Response to reviewer's comments

We thank the reviewers for their review and comments. We have revised our manuscript accordingly. Detailed below are our point-by-point responses.

Reviewer: 1

Comments to the Author(s)

The authors present a simple but comprehensive study comparing the effect of mutant proteins previously used to determine structural information by EPR on aggregation using ThT fluorescence. While it is a simple study the results are important for understanding of how spin labels may affect aggregation kinetics of Ure2 which could be important for future studies as suggested. I have some specific comments I feel should be addressed prior to publication of this manuscript to improve the relation of current findings to already known literature:

1. The article needs a conclusion section

Response: We have now added a conclusion section, which is copied below for the reviewer's convenience.

CONCLUSION

“Our results suggest that spin labeling in general has muted effect on the aggregation of Ure2. Only 5 out of 79 spin-labeled Ure2 variants showed markedly reduced aggregate rate. The clustering of these 5 residues in the region of 23-42 is consistent with previous findings that support the key structural role of the N-terminal half of the Ure2 prion domain. Interestingly, spin labeling at 6 out of 8 charged residue positions inhibited the rate of Ure2 aggregation, raising the prospect that the charged residues may serve as a molecular lubricant to provide necessary structural flexibility for fibril nucleation.”

2. The authors state in the introduction that R17C accelerates Ure2 under oxidising conditions implying this is a critical residue but then do not mention in the discussion how or if this relates to their current data.

Response: We have now included the work of R17C in the discussion (page 3, last paragraph). The relevant discussion is copied below.

“Previously, Perrett and coworkers identified that the R17C mutation accelerated Ure2 aggregation under oxidizing conditions and the peptide corresponding to the sequence of residues 17-24 can form amyloid fibrils in vitro [47].”

3. The buffer in the ThT reaction still contains a proportion of GdHCl (350mM) which should be stated and included in the figure legends.

Response: We have modified the statement regarding the buffer condition the legends of Figures 2, 3, and 4 to include this information. The relevant statement is as follows:

“The aggregation was performed using 10 μ M Ure2 in PBS buffer (pH 7.4, containing 0.35 M guanidine hydrochloride) at 37°C without agitation.”

4. Figure 4 would benefit from the mean being shown.

Response: The mean of half-time from all the mutants is now drawn in Figure 4, which becomes the new Figure 5 in the revised manuscript.

5. The authors discuss a dramatic difference, normally differences are discussed with relation to significance. I feel a specialist statistical reviewer should confirm if the statistical approach of taking average and 2SD (using n=2 or 3 per condition) is appropriate for this data rather than a more standard approach of comparison to WT protein values.

Response: With the aggregation kinetics of 79 mutants, we aim to focus on the main trend of the data rather than the smallest details. The vast majority of mutants show aggregation half-time that is close to WT protein. With only 2-3 repeats for every mutant, we do not have enough statistical power to distinguish if the difference between a mutant and wild-type is statistically significant or not. That is why we focused on the five residue positions (23, 27, 33, 35, and 42) that have half-time values two standard deviations larger than the mean of all mutants studied. This limits our discussion to a small number of mutants.

6. In the first paragraph of the results the authors refer to ThT as a quantitative measurement of amyloid fibrils but then later contradict this saying that ThT cannot be used as a quantitative measurement, this disparity should be removed.

Response: We thank the reviewer for pointing out this ambiguity. The discussion later on was meant to show that ThT fluorescence alone in mutagenesis studies cannot be used to quantify the effect of mutation. We revised the relevant paragraph and copied it below (Page 3, paragraph 3).

“Further analysis of the aggregation data of 79 Ure2 mutants suggests that site-specific effects on the rate of aggregation do not correlate with changes in thioflavin T fluorescence at aggregation plateau. Figure 7 shows a plot of thioflavin T fluorescence at aggregation plateau versus the half-time of aggregation for all the Ure2 mutants. It has been shown that thioflavin T fluorescence is a quantitative measure of fibril amount for the same fibril structure [48]. In mutagenesis studies of amyloid proteins, reduction in thioflavin T fluorescence has often been interpreted as inhibition of aggregation. A mutation can reduce the aggregation yield, but can also change the structure of the aggregate, which may bind thioflavin T with different affinity or give different fluorescence quantum yield even with the same binding affinity. Therefore, when

comparing mutants and wild-type aggregations, thioflavin T fluorescence intensity by itself cannot be used to quantify the effect of mutations on aggregation. We reached the same conclusion in a previous study of A β 42, in which we performed kinetics studies of 42 spin-labeled A β 42 mutants and found that the half-time of aggregation did not correlate with thioflavin T fluorescence at aggregation plateau [49].”

7. More details and discussion regarding the relevance of G42R1 being the only mutant to show lack of beta-sheet in previous EPR should be included and a discussion as to why the others may not cause a structural alteration with relation to currently presented findings.

Response: We added a new figure (Figure 6, copied below) to show the EPR spectra of the five mutants with the largest half-time values.

8. The authors comment on mutant positions that are near a turn (23, 42 and 33) however on their schematic position 33 appears to be within a beta-sheet region. This section should be re-written to fit with the schematic and further discussion of the implications of turn versus sheet positions.

Response: We have re-written this section in the discussion. This paragraph is copied below (page 3, paragraph 4).

“To help understand the aggregation data, we color-coded the residues in a schematic of Ure2 sequence that includes secondary structure information from our previous EPR studies (Figure 8). Out of the five most inhibiting residues, N23 and G42 are located at turn regions. S33 is located on a β -strand, but right adjacent to a turn. N27 and I35 are located in the middle of β -strands. The analysis suggests that formation of turns is as important as β -strands in the nucleation of fibrils, a key rate-limiting step in protein aggregation. Similarly, our previous kinetics studies of 42 spin-labeled A β 42 mutants showed that mutations at turn or loop sites had the most dramatic effect on A β 42 aggregation [49]. However, not all mutations at turns had the same effect, suggesting that there are site-specific interactions involved in fibril nucleation. Residue positions with the

most inhibiting effect do not appear to have a preference on residue type. These five residue positions include two glutamines, one isoleucine, one serine, and one glycine. Notably, the five residues do not have charged amino acids, even though there are 8 charged residues in the region of 2-80.”

Reviewer: 2

Comments to the Author(s)

In this manuscript, Liu and colleagues investigated how spine labelling on different regions of Ure2 effects its aggregation kinetics. The authors did a huge amount of work in preparing 79 spin-labelled Ure2 variants, respectively, which enables systematically comparison of all the variants of their aggregation kinetics. Overall, the work is technically sound, and is interesting and of importance to understanding the distinctive role of each individual residues on Ure2 in its fibrillation. I only have a few comments the authors might need to address in order to strengthen the manuscript.

1. The authors might need to provide EM images of at least the wild-type Ure2 sample in the end of the ThT kinetics assay. I am wondering whether Ure2 forms typical fibrils after incubation and what it looks like.

Response: We have included EM images of wild-type Ure2 in the new Figure 2 (copied below).

2. Does the spin-label influence the conformation of soluble Ure2?

Response: The prion domain of Ure2 is intrinsically disordered. Our previous EPR studies of 15 spin-labeled Ure2 mutants have shown that the spin-labeled Ure2 also adopts a completely disordered structure, suggesting that spin labeling does not affect the conformation of soluble Ure2. We have included this information in the Introduction. The relevant text is copied below (Page 2, paragraph 2).

“By immobilizing Ure2 protein on a solid-support, we demonstrated that monomeric Ure2 prion domain is completely disordered [32]. This is consistent with previous studies

that show the disordered nature of Ure2 prion domain [48], suggesting that spin labeling does not affect the conformation of soluble Ure2.”

3. The author might wish to discuss further whether the findings in this study can explain the previous observations for Ure2 aggregation and function in yeast?

Response: We have added two additional published studies (references 50 and 54) in our discussion. The relevant paragraph is copied below (Page 3, last paragraph).

“Previously, Perrett and coworkers identified that the R17C mutation accelerated Ure2 aggregation under oxidizing conditions and the peptide corresponding to the sequence of residues 17-24 can form amyloid fibrils in vitro [47]. Deletion of residues 15-41 dramatically reduced the rate of Ure2 aggregation in vitro [50]. Ventura and coworkers used the pWALTZ algorithm to predict and then experimentally validated that the 21-residue stretch spanning residues 20 to 40 of Ure2 formed an amyloid core [51]. Hydrogen exchange studies by Melki and coworkers [52] show that fibrillization gave the highest level of protection to the amide protons of residues 13-37. Solid-state NMR studies by Tycko and colleagues show that the peptide corresponding to residues 10-39 of the Ure2 prion domain formed fibrils with parallel in-register structure [53]. Wickner and colleagues show that residues 2-44 are important for prion induction and curing in vivo [54]. These studies support the key structural role of the N-terminal half of the Ure2 prion domain in the formation of Ure2 fibrils. Our findings that the five residues with most dramatic effect on Ure2 aggregation rate are clustered between residues 23 and 42 suggest that this region is also kinetically important for the aggregation of Ure2 prion domain.”